# Over-Expression of LEDGF/p75 in HEp-2 Cells Enhances Autoimmune IgG Response in Patients with Benign Prostatic Hyperplasia—A Novel Diagnostic Approach with Therapeutic Consequence?

**DOI:** 10.3390/ijms24076166

**Published:** 2023-03-24

**Authors:** Victoria Liedtke, Laura Rose, Rico Hiemann, Abdullah Nasser, Stefan Rödiger, Alena Bonaventura, Laura Winkler, Mandy Sowa, Michael Stöckle, Peter Schierack, Kerstin Junker, Dirk Roggenbuck

**Affiliations:** 1Faculty Environment and Natural Sciences, Brandenburg University of Technology Cottbus-Senftenberg, 01968 Senftenberg, Germany; victoria.liedtke@b-tu.de (V.L.);; 2GA Generic Assays GmbH, 15827 Blankenfelde-Mahlow, Germany; 3Medipan GmbH, 15827 Blankenfelde-Mahlow, Germany; 4Faculty of Health Sciences Brandenburg, Brandenburg University of Technology Cottbus-Senftenberg, 01968 Senftenberg, Germany; 5Department of Urology and Pediatric Urology, Saarland University, 66424 Homburg, Germany

**Keywords:** LEDGF/p75, autoimmunity, CRISPR/Cas9, dsDNA, mDNA

## Abstract

Lens epithelium-derived growth factor splice variant of 75 kDa (LEDGF/p75) is an autoantigen over-expressed in solid tumors and acts as a stress-related transcriptional co-activator. Participation of autoimmune responses in the pathophysiology of benign prostatic hyperplasia (PBH) and a corresponding immunosuppressive therapy by TNFalpha antagonists has been recently suggested. Thus, autoAb testing could aid in the diagnosis of BPH patients profiting from such therapy. We generated CRISPR/Cas9 modified HEp-2 LEDGF knock-out (KO) and HEp-2 LEDGF/p75 over-expressing (OE) cells and examined IgG autoantibody reactivity to LEDGF/p75 in patients with prostate cancer (PCa, n = 89), bladder cancer (BCa, n = 116), benign prostatic hyperplasia (BPH, n = 103), and blood donors (BD, n = 60) by indirect immunofluorescence assay (IFA). Surprisingly, we could not detect elevated binding of autoAbs against LEDGF/p75 in cancer patients, but autoAb reactivity to LEDGF/p75 OE cells in about 50% of patients with BPH was unexpectedly significantly increased. Furthermore, a line immunoassay enabling the detection of 18 different autoAbs revealed a significantly increased occurrence of anti-dsDNA autoAbs in 34% of BPH patients in contrast to tumor patients and BD. This finding was confirmed by anti-mitochondrial (mDNA) autoAb detection with the Crithidia luciliae immunofluorescence test, which also showed a significantly higher prevalence (34%) of anti-mDNA autoAbs in BPH. In summary, our study provided further evidence for the occurrence of autoimmune responses in BPH. Furthermore, LEDGF/p75 over-expression renders HEp-2 cells more autoantigenic and an ideal target for autoAb analysis in BPH with a potential therapy consequence.

## 1. Introduction

Lens epithelium-derived growth factor (LEDGF) splice variant of 75 kDa (LEDGF/p75), also known as transcriptional co-activator p75, PC4, and SFRS1 interacting protein (PSIP1), is a multifunctional stress-response protein acting as a transcriptional co-activator over-expressed in distinct tumor and inflammatory conditions [1]. The LEDGF/p75 protein is localized in condensed chromatin, interphase chromatin, and even perinucleolar chromatin, excluding the nuclei [2,3]. In recent years, LEDGF/p75 has emerged as an oncoprotein identified in multiple cancer types and shown to be over-expressed in various solid cancer types such as prostate (PCa), breast, thyroid, and colon cancer [4]. Over-expression of LEDGF/p75 promotes tumor aggressiveness and leads to increased proliferation, migration, chemoresistance, and augmented DNA repair [5,6,7].

Autoimmunity to LEDGF/p75 in the form of autoantibodies (autoAbs) against LEDGF/p75 has been shown in various diseases such as PCa as well as in healthy individuals [8,9,10,11]. In the context of the latter, antinuclear autoAbs (ANAs), demonstrating a dense fine speckled (DFS) nuclear pattern in indirect immunofluorescence assay (IFA) on human epithelial type 2 (HEp-2) cells and generally referred to as anti-DFS70 autoAbs, mainly interact with LEDGF/p75 [10]. Consistent with the international consensus on ANA patterns (ICAP), the DFS pattern encoded as AC2 belongs to the competent level in routine ANA testing (www.anapatterns.org, accessed on 24 November 2022). The detection of ANA is a screening tool for the serological diagnosis of systemic autoimmune rheumatic diseases (SARD), and ANA positivity has been introduced as an entry criterion for the classification of systemic lupus erythematosus recently [12]. Because of the low prevalence of anti-LEDGF (DFS70) autoAbs in patients with SARD (less than 1%) and high prevalence in healthy individuals (up to 22%), the assessment of these autoAbs has been suggested for reflex testing in the serological workup for patients with SARD to distinguish them from healthy individuals [13,14].

The sensitivity of anti-LEDGF/p75 autoAb detection appears to be dependent on the assay technique, and conflicting results about the prevalence of these autoAbs have been reported in tumor diseases, as reviewed elsewhere [2,10]. Therefore, we created LEDGF knock-out (LEDGF KO) HEp-2 cell clones as negative controls and corresponding clones over-expressing LEDGF/p75 and used these recombinant cells together with wild-type (WT) HEp2-cells in IFA for the detection of anti-LEDGF/p75 autoAbs. We aimed to minimize nonspecific autoAb binding, as suggested by Malyavantham and Suresh (2017) [15], and to increase the sensitivity of anti-LEDGF detection by IFA. For confirmation of anti-LEDGF/p75 autoAb positivity, we employed a multiplex line immunoassay (LIA) using human recombinant LEDGF/p75 as one of the solid-phase targets, as recommended elsewhere.

Surprisingly, we were not able to detect elevated autoAbs against LEDGF/p75 by IFA and LIA in patients with PCa or bladder cancer (BCa), but we observed a significant increase in autoAb reactivity against LEDGF/p75 over-expressing cells in sera of patients with benign prostatic hyperplasia (BPH). However, we could not confirm the presence of anti-LEDGF autoAbs by LIA in BPH patients but found elevated autoAbs to double-stranded (dsDNA) and mitochondrial DNA (mDNA) by LIA and IFA, respectively. Vickmann et al. C4 reported that autoimmune patients with concurrent BPH receiving immunosuppressive treatment with TNF antagonists demonstrated a reduced incidence of BPH [16]. Our study thus provided further evidence for the occurrence of autoimmune responses in BPH and for the potential that LEDGF/p75 over-expression renders HEp-2 cells more autoantigenic. Autoantibody analysis could aid in the diagnosis of BPH patients with an autoimmune component.

## 2. Results

### 2.1. LEDGF/p75 Over-Expression Increases autoAb Binding of Patients with BPH

To investigate the reactivity of IgG autoAbs to nuclear and cytoplasmic targets, which include LEDGF/p75 in particular, sera from 89 patients with PCa, 116 with BCa, and 103 with PBH as well as 60 BD were run on slides coated with HEp-2 WT, LEDGF/p75 KO, and LEDGF/p75 over-expressing cells in IFA. We evaluated processed slides using automated pattern recognition by Aklides and determined the fluorescence intensity and corresponding ANA pattern in case of autoAb presence in the sera. Surprisingly, qualitative and quantitative autoAb binding to LEDGF/p75 over-expressing HEp-2 cells of patients with BPH and not with tumor disease was significantly higher than to WT and KO HEp-2 cells (*p* < 0.05, respectively) (Figure 1 and Table 1). In contrast, such a significant difference was not observed for autoAb binding of sera from patients with PCa and BCa as well as BD (*p* > 0.5, respectively).

In addition, we determined a significantly increased qualitative and quantitative autoAb binding to LEDGF/p75 over-expressing HEp-2 cells in sera of BPH patients than in sera of patients with BCa and PCa as well as BD (*p* < 0.001, respectively) (Figure 1C). In contrast, there were no significant differences in the autoAb reactivity of patients and controls to LEDGF/p75 KO and WT cells (Figure 1 and Table 1).

### 2.2. LEDGF/p75 Over-Expression Increases the Frequency of Nucleolar and Speckled Patterns in Patients with BPH

Immunofluorescence analysis of all patient and BD sera on LEDGF/p75 over-expressing HEp-2 cells revealed a significantly increased percentage of the fine-speckled pattern (AC4) in contrast to IFA on WT and KO cells (*p* < 0.05, respectively) (Figure 2A). A similar constellation was observed with the nucleolar-clumpy pattern (AC9), which was also significantly more prevalent on LEDGF/p75 over-expressing cells (Figure 2A). Of 51 AC4-positive sera, 21 sera (41.2%) showed a mixed pattern with AC9. In contrast, we observed an expected prevalence (6.7%) of all patterns on WT HEp-2 cells in BD with no significant differences between the prevalences of the corresponding patterns on genetically modified and WT HEp-2 cells (*p* < 0.05, respectively) (Figure 2B). Except for the frequencies of AC4 and AC9 patterns, all remaining pattern frequencies were not significantly different from the corresponding fluorescence patterns on LEDGF/p75 KO and over-expressing HEp-2 cells of all patients (*p* > 0.5, respectively, Figure 2B). Apart from a significantly increased positive rate of the AC4 pattern in patients with PCa on HEp-2 WT cells (13.5%), in contrast to genetically modified HEp-2 cells, we established no further significant differences in patients with BCa and PCa (Figure 2B). In addition, we could not establish significantly different pattern prevalences when we compared the corresponding fluorescent patterns on the WT and genetically modified HEp-2 cells between patients with BCa and PCa as well as BD (*p* > 0.5, respectively, Figure 2B).

However, patients with BPH showed significantly more prevalent AC4 and AC9 patterns on LEDGF/p75 over-expressing HEp-2 cells than on KO and WT cells (*p* < 0.001, respectively, Figure 2B). Furthermore, the AC4 and AC9 pattern prevalences of BPH patients were significantly higher on LEDGF/p75 over-expressing HEp-2 cells than those of tumor patients and BD (*p* < 0.05, respectively). Of note, almost all AC9 patterns detected on LEDGF/p75 over-expressing HEp-2 cells (26/27, 96.3%) were from BPH patients, whereas only one PCA patient and no BCa patients or BD showed AC9 patterns in IFA.

Overall, only patients with BPH demonstrated a significantly elevated frequency of positive fluorescence patterns on LEDGF/p75 over-expressing HEp-2 cells (47.6%) than on KO and WT cells (Figure 3A).

### 2.3. Patients with BPH Show a Significantly Increased Prevalence of autoAbs against dsDNA in LIA

In light of the significant differences in autoAb binding and fluorescence patterns in patients and BD and, in particular, of the increased AC4 and AC9 prevalence on LEDGF/p75 overexpressed cells in BPH, we aimed to identify the distinct autoAb reactivities responsible for the positive IFA findings. To this end, we used a qualitative multiparametric LIA allowing the detection of autoAbs to 18 different autoantigenic targets encompassing LEDGF/p75, also referred to as DSF70 (Figure 3A). Only autoAbs to dsDNA revealed significantly different results between patients and BD, whereas there were significantly more positive results in patients with BPH and PCa than in patients with BCa and BD (*p* < 0.05, respectively) (Figure 3B and Table 1). In detail, patients with BPH demonstrated the highest prevalence of 34.0% (35/103), followed by patients with PCa (18.0%, 16/89) and patients with BCa (9.5%, 11/116) (Table 1).

### 2.4. Patients with BPH Showed autoAbs to mDNA Detected by CLIFT

To confirm the anti-dsDNA autoAb positivity by LIA, we run a qualitative IFA on Crithidia luciliae that is used for the specific analysis of autoAbs to mDNA. Anti-mDNA autoAb positivity in this assay format requires the staining of the kinetoplast that contains a distinct form of mDNA with unique epitope characteristics. As expected, only 1/60 sera of BD (1.7%) showed positive anti-mDNA autoAbs confirming the high specificity of CLIFT. However, patients with BPH demonstrated a significantly more prevalent occurrence of autoAbs to mDNA than patients with PCa and BCa as well as BD (35/103 vs. 5/89, 6/116, 1/60; *p* < 0.05, respectively) (Figure 3C and Table 1). In addition, significantly more anti-mDNA autoAb positive patients with BPH also had positive anti-dsDNA autoAbs in LIA (28/35, 80.0%) than anti-dsDNA autoAb positive patients with PCa (4/16, 25.0%) and BCa (3/11, 27.3%) as well as BD (0/2, 0%, *p* < 0.05, respectively) (Table 1). However, only 25.2% of all BPH cases (26/103) had positive autoAbs against mDNA as well as LEDGF/p75 overexpressing HEp-2 cells. Thus, 53.1% (26/49) of BPH patients with positive autoAbs against LEDGF/p75 over-expressing HEp-2 cells demonstrated positive anti-mDNA autoAbs.

## 3. Discussion

The role of LEDGF/p75, also referred to as PSIP1 as an autoantigenic target in tumorigenesis, is yet poorly understood [10]. In the context of loss of humoral tolerance against LEDGF/p75, we, therefore, aimed to investigate the occurrence of autoAbs to this oncogenic and survival antigen in serum samples from cancer patients as reported elsewhere [1,2,10]. To this end, we generated a HEp-2 cell line over-expressing LEDGF/p75 and used it together with WT and KO HEp-2 cell lines as targets for autoAb analysis. Surprisingly, we could not detect autoreactivity to LEDGF/p75 in serum samples from cancer patients using LEDGF/p75 over-expressing HEp-2 cells in IFA and recombinant LEDGF/p75 in LIA, as reported elsewhere [1,17,18]. In contrast, we detected an elevated autoAb reactivity on LEDGF/p75 over-expressing HEp-2 cells in IFA in about 50% of patients with BPH for the first time. To the best of our knowledge, this is the highest incidence of autoAb positivity in BPH reported to date. BPH is the most common disease in men over 50 years of age, with a poorly understood pathogenesis [19]. One-third of BPH patients have no improvement in lower urinary tract symptoms with the current therapy (5alpha-reductase inhibitors, alpha-adrenergic receptor antagonists) or display recurrence of disease after short-term improvement [16]. When associated with severe symptoms, BPH can require surgical removal of the affected tissue. Therefore, early detection of patients at risk with an autoimmune component could have a positive impact on disease progression and provide further treatment options.

Thus, on the one hand, our data corroborate the findings of Bizzaro et al. [20] and Mahler et al. [9], who detected very low frequencies of anti-LEDGF/p75 autoAbs (DFS70) by IFA (<2%) or none at all by chemiluminescence immunoassay in tumor patients [9,20]. On the other hand, our data about the occurrence of autoAb binding to nuclear targets of LEDGF/p75 over-expressing HEp-2 cells in BPH support an autoimmune component in the pathophysiology of BPH. There is mounting evidence that BPH can be considered an autoimmune disease or at least be closely associated with autoimmunity [21]. Interestingly, TNF-antagonists used for the therapy of autoimmune disorders may be viable therapeutics to reduce BPH incidence in patients with autoimmune diseases and decrease localized inflammation within the prostate [16]. Thus, autoAbs to prostate targets could have a diagnostic or even pathogenic role for BPH [22,23,24,25]. Patients with BPH produced anti-prostate-specific antigen (PSA) autoAbs, whereas limited to no anti-PSA autoAbs were found in patients with PCa or prostatitis [24]. Furthermore, there is emerging evidence that men with BPH can develop a dysregulated immune response via elevated expression of IL-7, which in turn increases the expression of IL-6 and IL-8, both key regulators of stromal growth of BPH [21]. Overexpression of LEDGF/p75 in Hep-2 cells appears to render these cells more autoantigenic for BPH patients. This phenomenon could be due to its transcriptional co-activator function and corresponding interaction with the DNA, which could result in the formation of neoepitopes. Enhanced oxidative stress but no obvious oxidative damage has been recently observed in a BPH rat model with autoimmune prostatitis [26]. We speculate that LEDGF/p75 is expressed at a higher level under such circumstances and triggers the loss of tolerance, which in turn perpetuates the inflammatory process in the prostatic tissue.

As possible targets of autoAbs to LEDGF/p75 over-expressing Hep-2 cells in our study did not appear to be prostate-specific, we sought to identify the autoantigens responsible for the most frequently detected ANA patterns AC4 (speckled) and AC9 (nucleolar-clumpy) in BPH. The occurrence of AC4 and AC9 is consistent with a report by Wichainun et al., who showed the association of speckled and nucleolar ANA patterns in a patient with BPH Investigating healthy individuals and patients with multiple medical problems [27]. Of note, the detection of ANA on hEp-2 cells has become an important screening tool in the serological diagnosis of SARD (entry criterion for SLE) and has been recognized as the gold standard for ANA testing [12,28]. In addition, ANA is a classification criterion for autoimmune hepatitis [29]. However, although ANAs are important markers of autoimmune disease, they are not unique to autoimmune disorders, as multiple studies report the involvement of ANAs in a variety of neoplastic diseases [30].

To this end, we used a multiplex LIA that enabled the detection of 18 autoAbs commonly used for the serological diagnosis of SARD and autoimmune liver diseases. Notably, this LIA also included the detection of autoAbs to recombinant LEDGF/p75. However, we again confirmed the above-mentioned low frequency of anti-LEDGF/p75 autoAbs in pCa patients but did also not ascertain significant autoAb levels in patients with BPH and bCa as well as BD. As expected and reported elsewhere, the latter demonstrated the highest prevalence (5.4%), although not significantly different from those of the patient cohorts [9,13]. In contrast, about one-third of patients with BPH demonstrated autoAbs to dsDNA by LIA. Because dsDNA-AutoAbs generally produce homogeneous patterns on hEp-2 cells with positive staining of metaphase chromatin and we could not detect such a pattern (AC1) on the LEDGF/p75-overexpressing hEp-2 cells, but rather AC4 and AC9, we attempted to confirm the dsDNA autoAb reactivity by another method such as CLIFT. The latter method uses the hemoflagellate parasite Crithidia luciliae and, in particular, its kinetoplast DNA as a substrate for autoAb assessment by IFA [31,32]. The kinetoplast represents a uniquely large mitochondrion that contains a circular mDNA characterized by hypomethylated CpC motifs [33]. Anti-mDNA autoAbs detected by CLIFT and by ELISA have been reported to be associated with autoimmune conditions that encompass SLE, and in particular with disease severity [31,34,35]. In contrast to BD and tumor patients, patients with BPH demonstrated a high level of consistency of anti-dsDNA and anti-mDNA autoAbs detected by LIA and CLIFT in our study, respectively. Remarkably, none of these BPH patients concurrently suffered from an autoimmune disorder and especially not from SLE. This fact and the anti-mDNA positive rate in CLIFT of 34.0% suggest a humoral loss of tolerance against a particular form of DNA found in mitochondria in BPH. These anti-mDNA autoAbs of the IgG isotype might be responsible for the unusual patterns (AC4 and AC9) on LEDGF/p75 over-expressing HEp-2 cells. LEDGF/p75’s perinucleolar location could result in molecular changes in the nucleoli in the event of overexpression of the molecule. Of note, LEDGF/p75 itself has not been found in nucleoli to date, and the nucleolar-clumpy pattern (AC9) was almost exclusively detected in BPH patients in this study [2]. Notwithstanding, only 53.1% of BPH patients positive on LEDGF/p75 over-expressing HEp-2 cells exhibited anti-mDNA autoAbs. Given the high consistency of anti-mDNA with anti-dsDNA reactivity in BPH patients, it is very likely that other molecules could serve as neoepitopes.

Our study is not without limitations. It does not include a verification cohort of BPH patients and therefore needs to be substantiated by further studies. Furthermore, dsDNA and mDNA do not appear to be the only autoantigenic targets recognized by autoAbs in BPH patients. Further studies are needed to identify these BPH-specific targets and shed more light on the loss of tolerance in BPH.

In summary, ANA to LEDGF/p75 over-expressing HEp-2 cells generated by CRISPR/Cas9 and anti-mDNA autoAbs by CLIFT could aid in the identification of BPH patients with an autoimmune involvement in combination with other prostate-specific autoAbs. The specific occurrence of these autoAbs supports an autoimmune component in the pathophysiology of the disease and could be a diagnostic option for immunosuppressive therapy.

## 4. Materials and Methods

### 4.1. Patients

In total, 308 patients suffering from PCa (n = 89), BCa (n = 116), and BPH (n = 103), as well as 60 (52 male/8 female) healthy blood donors (BD), were enrolled in the study (Table 2). Only 2 of 103 patients with BPH demonstrated a concomitant autoimmune disease (Hashimoto’s thyroiditis, psoriasis exantherica with mesangioproliferative glomerulonephritis).

Table 2 shows the characteristics of all analyzed patients, their age and age range, interquartile range (IQR), gender, and percentage of males in the corresponding cohort. Tumor stages in patients with bladder cancer are either not available (N/A) or range from noninvasive (pTa) to tumor that has spread into the stroma of the prostate (man) and to the uterus and/or vagina (woman) (pT4). In patients with prostate cancer tumor, stages range from a tumor that has grown into the inner half of the muscle layer (pT2a) to a tumor that has grown into the fatty tissue and can be seen on imaging tests or can be felt by the surgeon (pT3b) [36].

### 4.2. Generation of LEDGF-Modified Cell Clones

The generation of modified cell lines has been described elsewhere [7]. Briefly, HEp-2 WT cells were transfected with px458_sgR_DFS70_E1 using LipofectamineTM 3000 according to the manufacturer’s instructions (Thermo Fisher Scientific, Massachusetts, MA, USA). For LEDGF/p75 over-expressing, WT and KO HEp-2 cells were co-transfected with px458_sgRNA_AAVS1 and pAAVS1_CAG-EGFP-LEDGF/p75. Transfected cells were enriched by EGFP selection of biomarkers via FACS using an S3e cell sorter (Bio-Rad). Successfully transfected cells were sorted by GFP expression, and per 10 cm^2^ cell culture plate, a total of 1 × 103 cells were seeded. Outgrown fluorescent, single-cell colonies were picked after 7–10 days to establish LEDGF o/e cell lines. Subsequently, the cell clones were analyzed to verify the integration of the expression cassette at the AAVS1 locus.

### 4.3. Cell Lines and Culture

HEp-2 LEDGF/p75 wild-type (WT) cells and LEDGF/p75 knock-out (KO), as well as LEDGF/p75 over-expressing (LEDGF/p75) cells, were grown up to 80% confluence in DMEM/Ham’s F12 supplemented with 10% FBS (Biowest, Nuaillé, France), 2 mM L-glutamine (Merck Millipore, Massachusetts, MA, USA), and 1x penicillin/streptomycin (Merck Millipore, Massachusetts, MA, USA) in a humidified incubator at 37 °C and 5% CO_2_. WT and LEDGF/p75 cell lines were split in a 1:10 ratio and LEDGF KO cells in a 1:5 ratio.

### 4.4. Detection of autoAbs by IFA

For autoAb analysis, WT and recombinantly modified HEp-2 cells were seeded at 5 × 10^3^ cells/well on 12-well slides (GA Generic Assays GmbH, Dahlewitz, Germany) and incubated for 24 h. For analysis, cells were fixed according to the in-house protocol (GA Generic Assays GmbH) and dried at room temperature. Serum samples (diluted in PBS 1:80) and controls were added and incubated at RT for 1 h. Slides were washed 3 times for 5 min with PBS, then incubated with secondary antibody (1:500 Anti-human IgG 647 [GA Generic Assays, Germany]) and DAPI (5 µg/mL [VWR, Pennsylvania, PA, USA]) for 1 h in the dark at RT. Fluorophore photostability was increased by coating slides with a mounting medium (Roti^®^-Mount FluorCare, Carl Roth GmbH, Karlsruhe, Germany). Analysis was performed using a confocal laser scanning microscope LSM 800 (Zeiss, Oberkochen, Germany) and the fully-automated image detection system AKLIDES (Medipan, Dahlewitz, Berlin, Germany) [37,38].

### 4.5. Detection of autoAbs by Line Immunoassay

For the simultaneous analysis of 18 autoAbs, a multiparameter LIA was used in accordance with the manufacturer’s instruction (ANA18, GA Generic Assays GmbH). Briefly, nitrocellulose was coated with dsDNA, nucleosome, Smith antigen (Sm), Sm/ribonucleoprotein (RNP), ribosomal protein 0 (P0), histone, U1-small nuclear RNP (U1-snRNP), Sjögren’s syndrome antigen A (SS-A)/Ro60, SS-A/Ro52/tripartite motif-containing protein 21 (TRIM21), Sjögren’s syndrome antigen B (SS-B/La), scleroderma antigen 70 kDa (Scl-70), polymyositis and scleroderma antigen 100 kDa (PMScl-100), centromere proteins A and B (CENP-A/B), proliferating cell nuclear antigen (PCNA), Jo-1, anti-mitochondrial antibody antigen M2 (AMA-M2), DFS70 (LEDGF/p75), and filamentous actin (f-actin). All reagents and required test strips were brought to room temperature before use. Test strips were placed in an incubation tray and incubated with 1.5 mL sample diluent for 5–10 min at RT on a shaker. Subsequently, 10 µL of patient serum was added directly to each test strip in the incubation tray and incubated for 30 min at RT on a shaker. Sample diluent and serum were decanted, and test strips were washed 3 times for 3 min with washing solution, followed by incubation of each test strip with 1.5 mL conjugate solution (anti-human IgG conjugated to horseradish peroxidase) for 30 min at RT on a shaker. After washing 3 times for 3 min, strips were incubated with 1.5 mL substrate TMB solution for 10–12 min at RT on a shaker. Subsequently, the reaction was stopped by a final washing step, and test strips were dried on an absorbent pad for analysis of results after 20 min.

### 4.6. Analysis of autoAbs to mDNA by IFA

For the detection of IgG autoAbs to mDNA, an IFA with Crithidia luciliae mDNA as a solid-phase autoantigenic target (CLIFT) was used (GA Generic Assays GmbH) [31,35]. All incubation steps were performed according to the manufacturer’s instructions. Briefly, controls and diluted patient sera (1:100 in sample diluent) were pipetted directly onto the delivered slides with fixed Crithidia luciliae and incubated for 30 min at RT. Afterwards, slides were washed 3 times for 5 min with PBS, followed by incubation with conjugate solution (25 µL) for 30 min at RT. Slides were washed 3 times again for 5 min in 1× PBS, the mounting medium was added to the wells, and the slide was covered with a coverslip. Analysis was performed using a confocal laser scanning microscope LSM 800 (Zeiss, Oberkochen, Germany).

### 4.7. Statistical Analysis

All data were statistically analyzed with the statistical computing language R v. 3.6 [39]. Fisher’s exact test was used to analyze contingency tables, and the Kruskal–Wallis test was employed to compare cohorts. To control the α error inflation, the Bonferroni correction was applied. Tukey’s HSD test was used to test the differences between the mean values of the sample for significance. The cut-off for the IFA analyses was determined using the 3-sigma limit based on the averaged negative controls for the assay; *p*-values less than 0.05 were considered significant. Experiments were conducted with at least three replicates.

## Figures and Tables

**Figure 1 ijms-24-06166-f001:**
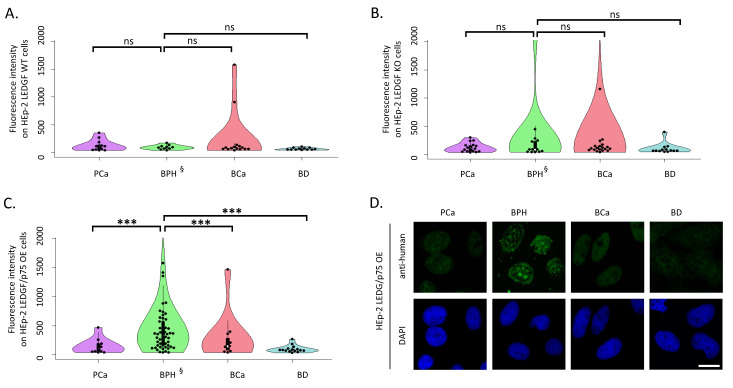
Autoantibody binding of serum from patients and blood donors (BD) on HEp-2 wild-type (WT), LEDGF/p75 knock-out (KO), and LEDGF/p75 over-expressing (OE) cells detected by indirect immunofluorescence assay. Fluorescence intensity was measured by AKLIDES software after incubation of sera from 60 BD, patients with prostate cancer (PCa) (n = 89), with bladder cancer (BCa) (n = 116), and with benign prostate hyperplasia (BPH) (n = 103) on A. HEp-2 WT cells, B. HEp-2 LEDGF KO cells, and C. HEp-2 LEDGF/p75 OE cells (pPCa-BPH = 1 × 10^−7^, pBPH-BCa = 9 × 10^−4^, pBPH-BD = 9 × 10^−5^), followed by incubation of anti-human IgG secondary antibody. Single dots represent the value of a single human serum, and fluorescence intensity showed the intensity of the obtained ANA pattern measured by AKLIDES. § statistical difference between the reactivity of BPH patients on (**A**–**C**), *p* < 0.001, respectively. (**D**) Representative immunofluorescence patterns of different serum types, scale bar = 10 µm. *** *p* < 0.001; ns *p* > 0.05.

**Figure 2 ijms-24-06166-f002:**
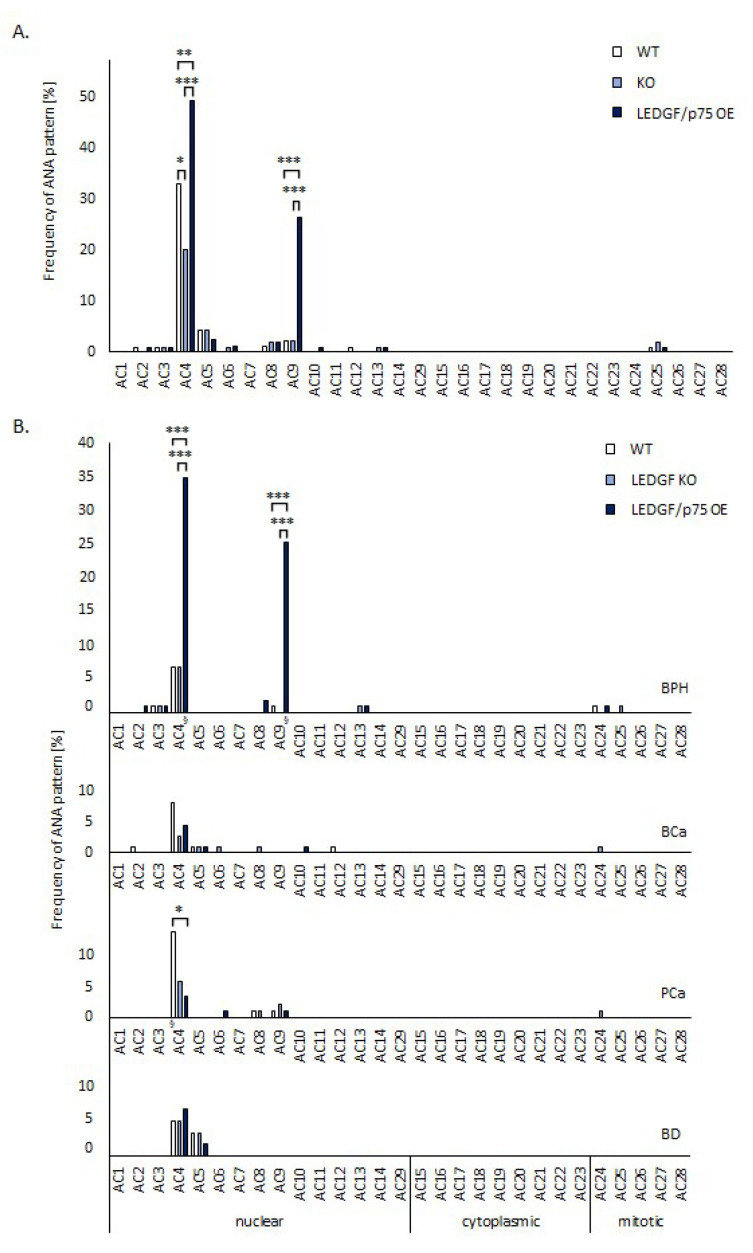
Fluorescence patterns on HEp-2 wild-type (WT), LEDGF/p75 knock-out (KO), and LEDGF/p75 over-expressing (OE) cells. Sera of 60 blood donors (BD), patients with prostate cancer (PCa) (n = 89), with bladder cancer (BCa) (n = 116), and with benign prostate hyperplasia (BPH) (n = 103) were run on wild-type (WT) and recombinantly modified HEp-2 cells. (**A**) Overall frequency in percent (%) of fluorescence ANA--pattern detected on HEp-2 WT, LEDGF KO, and LEDGF/p75 OE HEp-2 cells after incubation with mentioned serum types according to ICAP criteria. Statistical significance was determined by Fisher’s exact test. (**B**) Frequency in percent (%) of ANA patterns according to ICAP criteria detected on HEp-2 WT, LEDGF KO, and LEDGF/p75 OE HEp-2 cells detected in patients and BD. * = *p* < 0.05, ** = *p* < 0.01, *** = *p* < 0.001; § comparison of BPH patients’ reactivity on LEDGF/p75 OE HEp-2 cells with those of tumor patients and BD, *p* < 0.001, respectively.

**Figure 3 ijms-24-06166-f003:**
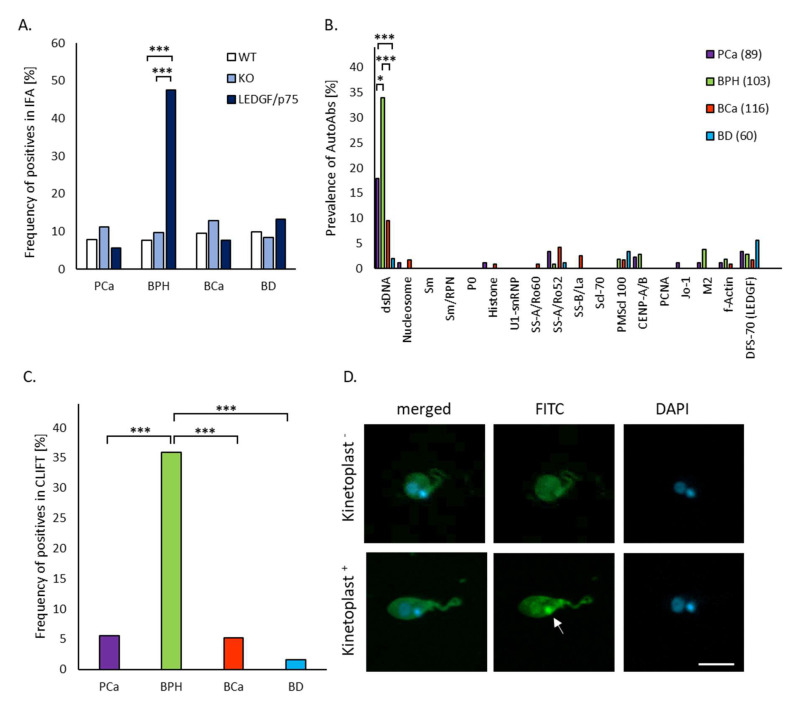
Autoantibodies (autoAbs) in patients and blood donors (BD) detected by indirect immunofluorescence assay (IFA) on wild-type (WT), LEDGF/p75 knock-out (KO), LEDGF/p75 over-expressing (OE) HEp-2 cells, and line immunoassay (LIA) as well as Crithidia luciliae immunofluorescence test (CLIFT). Sera from 60 BD, patients with prostate cancer (PCa) (n = 89), with benign prostate hyperplasia (BPH) (n = 103), and bladder cancer (BCa) (n = 116) were run (**A**). In IFA to analyze the overall rate of positives on WT, LEDGF/p75 KO, and LEDGF/p75 OE HEp-2 cells, (**B**) In LIA to detect autoAbs to 18 distinct autoantigens and (**C**). In IFA on Crithidia luciliae to ascertain autoAbs to mDNA. (**D**) Representative fluorescent images of anti-mitochondrial DNA (mDNA) autoAb-negative and positive sera in IFA using DAPI (blue color) to stain DNA and anti-human IgG-FITC (green color) to identify the binding of autoAbs. A positive anti-mDNA autoAb finding requires the staining of the kinetoplast (arrow) (kinetoplast+), which contains mDNA with distinct epitopes not present in the nucleus. Scale bar = 10 µm. * = *p* < 0.05, *** = *p* < 0.001.

**Table 1 ijms-24-06166-t001:** Qualitative autoantibody (autoAb) results from patients with bladder cancer (BCa), benign prostatic hyperplasia (BPH), and prostate cancer (PCa), as well as blood doors (BD). IgG autoAbs against HEp-2 wild-type (WT), LEDGF/p75 knock-out (KO), and LEDGF/p75 over-expressing (OE) HEp-2 cells, double-stranded (dsDNA), and mitochondrial DNA (mDNA) were ascertained by indirect immunofluorescence assay, line immunoassay, and Crithidia luciliae immunofluorescence test, respectively.

autoAbs to	BPH (103) n (%)	BCa (116) n (%)	PCa (89) n (%)	BD (60) n (%)
**LEDGF/p75 HEp-2**	49 (47.6%) *§	9 (7.8%)	5 (5.6%)	5 (8.3%)
**LEDGF KO HEp-2**	10 (9.7%)	15 (13.0%)	10 (11.2%)	5 (8.3%)
**LEDGF WT HEp-2**	8 (7.8%)	11 (9.5%)	7 (7.9%)	5 (8.3%)
**dsDNA**	35 (34.0%) *	11 (9.5%)	16 (18.0%)	2 (3.3%)
**mDNA**	35 (34.0%) *	6 (5.2%)	5 (5.6%)	1 (1.7%)
**LEDGF/p75 HEp-2 + dsDNA**	23 (22.3%) *	1 (0.9%)	1 (1.1%)	0 (0.0%)
**LEDGF/p75 HEp-2 + mDNA**	26 (25.2%) *	1 (0.9%)	1 (1.1%)	0 (0.0%)
**dsDNA + mDNA**	28 (27.2%) *	3 (2.6%)	4 (4.5%)	0 (0.0%)

* = *p* < 0.05 comparison to BD. § comparison of BPH patients’ reactivity on LEDGF/p75 OE HEp-2 cells with LEDGF WT cells, *p* < 0.001, respectively.

**Table 2 ijms-24-06166-t002:** Patients and blood donor cohort characteristics.

	Age (Median) [years]	Age Range [years]	Interquartile Range (IQR) [years]	Gender	Male [%]	Tumour Stage	Gleason Score
**BPH**	70.0	50–88	14	Male	100%	N/A	N/A
**BCa**	77.2	50–96	N/A	female + male	81%	pTa-pT4	N/A
**PCa**	64.0	43–77	N/A	male	100%	pT2a-pT3b	7a-9b
**BD**	34.0	21–58	N/A	female + male	86.7%	-	-

## Data Availability

Data is contained within the article.

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
