# Peer review of "Over-Expression of LEDGF/p75 in HEp-2 Cells Enhances Autoimmune IgG Response in Patients with Benign Prostatic Hyperplasia—A Novel Diagnostic Approach with Therapeutic Consequence?"

_ijms, 2023, doi:10.3390/ijms24076166_

Round 1

Reviewer 1 Report

In the manuscript of V. Liedtke et al. “Over-expression of LEDGF/p75 in HEp-2 cells enhances auto-2 immune IgG response in patients with benign prostatic hyper-3 plasia - a novel diagnostic approach with therapeutic conse-4 quence?” evidence of the occurrence of autoantibodies and autoimmune responses in patients with prostatic hyperplasia (BPH) is presented in comparison with control groups of patients with tumors and healthy blood donors (BD). The object of study is very complex; it all depends on the patients and their state at that time of blood sampling. Nevertheless, the manuscript is interesting; the data obtained can be used in the diagnosis of the disease, as well as in choosing the optimal treatment strategy. In general, the article is well written, contains interesting data. At the same time, it is performed at a good technical level. I suggest that after minor revisions, the manuscript may be published in IJMS. I have a couple of comments on the pictures:

In Figure 2, I recommend changing the color, it is difficult to distinguish what belongs to LEDGF / p75 OE in the diagram, and what to WT.

In Figure 3B, there is no label for the y-axis.

Author Response

Response to Reviewer 1 Comments

We would like to thank the Reviewer for the positive comment and respond as follows:

Point 1.1. In Figure 2, I recommend changing the color, it is difficult to distinguish what belongs to LEDGF / p75 OE in the diagram, and what to WT.

Response 1.1: Thank you for your recommendation. We have changed the colors in figure 2 and 3A to white, light blue and dark blue hoping to have achieved a better differentiation between the individual cell lines.

Point 1.2: In Figure 3B, there is no label for the y-axis

Response 1.2: Thanks also for this - the diagram label of the y-axis in Figure 3B has now been added.

Reviewer 2 Report

In this research,  the authors performed a study about detecting autoantibodies in the sera of benign prostate hyperplasia patients (BPH). The LEDGF/p75 was selected as an autoantigen, and the  autoantibodies to the antigen was tested using sera of BPH and the sera of prostate cancer, bladder cancer, blood donor were used as controls. They also applied a line immunoassay with 18 autoantibodies. The results were consistent with previous knowledge and has an clinical impact. I'd like to address some minor issues. 

1. Figures should be readjusted. I think the figures have resolution problems. Especially bar widths in the figure 2 should be enlarged, and colors reassigned to combinations that are more easily discriminated between the AC groups. 

2. In the introduction, the following phrase should be checked. "Im-80 munosuppressive treatment of autoimmune patients with concurrent BPH by TNF antag-81 onists has been recently shown to reduce the incidence of BPH in these patients.  Is it correct to insert "with concurrent BPH" into "autoimmune patients with concurrent BPH by TNF antag-81 onists "?

3. In figure 2, LEDGF KO and OE showed the same frequency in the AC4. Authors should provide possible explanation about this finding. 

Author Response

Response to Reviewer 2 Comments

We would like to thank the Reviewer for the positive comment and respond as follows:

Point 1.1: Figures should be readjusted. I think the figures have resolution problems. Especially bar widths in the figure 2 should be enlarged, and colors reassigned to combinations that are more easily discriminated between the AC groups. 

Response 1.1: Thank you for your helpful remark. We have increased the bar width from 0.5 to 1 pt  and changed the colors in Figure 2 and 3A to white, light blue and dark blue hoping to have achieved a better discrimination of the individual cell lines. Because of the resolution problems, the images were uploaded again in a high-resolution format.

Point 2.2: In the introduction, the following phrase should be checked. "Im-80 munosuppressive treatment of autoimmune patients with concurrent BPH by TNF antag-81 onists has been recently shown to reduce the incidence of BPH in these patients.  Is it correct to insert "with concurrent BPH" into "autoimmune patients with concurrent BPH by TNF antag-81 onists "?

Response 2.2:  

We appreciate the Reviewer’s concern regarding our statement. We rephrased the sentence for better understanding:

“Vickmann et al. reported that autoimmune patients receiving immunosuppressive treatment with TNF antagonists demonstrated a reduced incidence of BPH [16].”  

Point 3.3: In figure 2, LEDGF KO and OE showed the same frequency in the AC4. Authors should provide possible explanation about this finding. 

Response 2.3:

We thank the Reviewer for this request. We asscume that Fig. 2B was meant as the frequencies for AC4 on LEDGF/p57 knock-out (KO) and overexpressing (OV) cells were significantly different in patients and controls. With regard to Fig. 2B, patients with benign protatic hyperplasia (BPH) did show significant differences in the frequency of th AC4 patterm. Therefore, we assume that the Reviewer focused on tumor patients and blood donors as these individuals did not demonstrate significant differences in AC4 frequencies. Apart from patients with systemic autoimmune rheumatic diseases such as systemic lupus erythematosus and Sjogren’s syndrome, the ICAP pattern AC4 might be relatively frequent in healthy individuals (Mariz et al. Pattern on the antinuclear antibody-HEp-2 test is a critical parameter for discriminating antinuclear antibody-positive healthy individuals and patients with autoimmune rheumatic diseases. ARD 2011;63:191-200). Since tumor patients and BD did not show significant differences on the wild-type and genetically modified cells, we speculate that overexpression or knock-out of LEDGF/p75 did not alter the autoantigenicity of the respective cells in respect to these individuals. We hope we have adequately answered the reviewer's question.